# Cannabidiol (CBD) Dosing: Plasma Pharmacokinetics and Effects on Accumulation in Skeletal Muscle, Liver and Adipose Tissue

**DOI:** 10.3390/nu14102101

**Published:** 2022-05-18

**Authors:** Robert B. Child, Mark J. Tallon

**Affiliations:** 1School of Chemical Engineering, University of Birmingham, Birmingham B15 2TT, UK; 2Legal Products Group Ltd., 26 St Thomas Pl, Ely, Cambridgeshire CB7 4EX, UK; drtallon@legalproductsgroup.com

**Keywords:** CBD, cannabidiol, cannabis, cannabinoids, pharmacokinetics, muscle, liver, adipose, fat, metabolism

## Abstract

Oral cannabidiol (CBD) consumption is widespread in North America and Europe, as it has analgesic, neuroprotective and antitumor effects. Although oral CBD consumption in humans affords beneficial effects in epileptic and inflammatory states, its pharmacokinetics and subsequent uptake into tissue are largely unknown. This study investigated plasma pharmacokinetics and accumulation of CBD in gastrocnemius muscle, liver and adipose tissue in adult rats following oral gavage. CBD was fed relative to body mass at 0 (control), 30, 115, or 230 mg/Kg/day for 28 days; with 6 males and 6 females per dosing group. Pharmacokinetics were assessed on day 1 and day 28 in the group receiving CBD at 115 mg/Kg/day. The rise in tissue CBD was closely related to specific pharmacokinetic parameters, and adipose tissue levels were ~10 to ~100 fold greater than liver or muscle. Tissue CBD levels were moderately correlated between adipose and muscle, and adipose and liver, but were highly correlated for liver and muscle. CBD feeding resulted in several gender-specific effects, including changes in pharmacokinetics, relationships between pharmacokinetic parameters and tissue CBD and differences in tissue CBD levels. CBD accumulation in mammalian tissues has the potential to influence receptor binding and metabolism; therefore, the present findings may have relevance for developing oral dosing regimens.

## 1. Introduction

Cannabidiol (CBD) or 2-[(6R)-6-isopropenyl-3-methyl-2-cyclohexen-1-yl]-5-pentyl-1,3-benzene-diol) is naturally present in *Cannabis sativa* L. and is one of 500 compounds and 100 cannabinoids identified in cannabis [1]. CBD has a tetrahydrobiphenyl skeleton: a bicyclic core and is an adduct formed by the monoterpene, p-cymene and the alkylresorcinol derivative, olivetol [2]. Of the phytogenic cannabinoids discovered to date, CBD is considered atypical because of its promising effects in a wide variety of diseases [3]. Importantly, although CBD is psychoactive [4], unlike Tetrahydrocannabinol (THC), it is not psychotogenic [5,6]. These unusual characteristics make CBD particularly relevant for drug development [3].

Tetrahydrocannabinol (THC) has been the traditional focus of cannabinoid research; however, recent in vitro and in vivo studies highlight CBD’s potential in an increasing range of therapeutic applications. CBD has anti-inflammatory, analgesic, anticonvulsant, anxiolytic, anti-epileptic, neuroprotective and antitumor properties [7]. It is thought to act via diverse molecular targets, including G protein coupled receptors and the cannabinoid receptors CB1 and CB2. Receptors for serotonin, adenosine and opioids are also obvious targets for CBD action and are widely distributed throughout the bodies of mammals [3].

Despite the recognised benefits of CBD administration, there are considerable knowledge gaps regarding its pharmacokinetics and metabolism [8]. For example, many human and animal studies administered CBD intravenously [9], which limits their relevance to oral consumption. These routes of administration bypass absorption by the lymphatic system and increase hepatic first pass extraction [10]. Other human trials, though providing CBD orally, also co-administered THC [11,12,13]. This has a similar structure [14] and lipophilicty (log *P* = 5.91) [15] to CBD and these characteristics affect CBD pharmacokinetics and metabolism [16]. These effects are partially due to similarities in the time course for plasma appearance [17] and receptor binding [16]. Research on plasma CBD pharmacokinetics may have practical value for determining optimal oral dose–response effects to maximise tissue accumulation. For example, it has recently been proposed that peak plasma concentration (C_max_) may correlate with the pharmacological effect of a substance, while the time to peak plasma concentration (t_max_) may relate to the onset of a substance’s action [18]. The effects of long-term feeding of CBD on tissue accumulation are also poorly understood, as most studies have focused solely on THC [19] or co-consumption of THC and CBD from cannabis [20]. The amount of CBD that accumulates in tissue is likely to be more important than pharmacokinetics for exerting its well-documented effects on membrane channels, receptors and enzymes [3].

The present investigation was performed on male and female adult rats to give easy access to relevant tissues and provide novel insights into mammalian CBD metabolism. Three physiologically relevant oral CBD doses were chosen to investigate its dose response accumulation in muscle, liver and adipose tissue. In addition, plasma CBD pharmacokinetics were evaluated in response to acute and chronic intake. This approach allows for the relationship between plasma pharmacokinetics and tissue accumulation to be determined, providing novel insights that are relevant to human research on CBD safety and efficacy.

## 2. Materials and Methods

### 2.1. Animals

All experiments were performed in accordance with the Guide for the Care and Use of Laboratory Animals [21].

Animals were 48 Sprague–Dawley CD IGS rats—comprising 24 males and 24 females—which were nulliparous and non-pregnant. All rats were 7 to 8 weeks old and were considered to be adults upon initiation of the trial. On commencing the trial, the variation in body mass was less than ± 20% of the mean mass for each gender. The rats were housed in individual cages with a 12-h light–dark cycle. Temperature and humidity were set at 21 ± 2 °C and 30–70%, respectively, with 10 air changes per hour. Animals were familiarised with the housing facility for at least five days before starting experimental procedures. Tap water and food (Certified Envigo Teklad Global Rodent Diet, Envigo Teklad Inc, Indianapolis, IN, USA) were both provided ad libitum during the familarisation period and experimental trial. Animals were fasted for 15 h prior to pharmacokinetic investigations.

### 2.2. CBD Characterisation and Administration

The stock CBD for dosing was supplied by cbdMD (manufacturer, Charlotte, NC, USA) and comprised 32.95% CBD in medium chain triglyceride oil (Batch No. 02751). The CBD concentration was independently determined by Liquid Chromatography–Diode Array Detection [22] against a reference standard (Sigma-Aldrich, Saint Louis, MI, USA). The CBD stock was stored at ambient conditions in the dark and prepared on the day of feeding by mixing with medium chain triglycerides (MCT). Each CBD dosing was used within 2 h of preparation and maintained on a stir plate up to the point of administration.

Animals were randomly assigned to four dosing groups, each comprised of 6 males and 6 females. The stock CBD was freshly diluted each day into a medium chain triglyceride carrier and delivered in a volume of 5 mL/kg via oral gavage. Each group was fed CBD relative to body mass at doses of 0 mg/kg/day as a control comprised solely of the CBD carrier, 30 mg/kg/day (“low dose”); 115 mg/kg/day (“medium dose”) and 230 mg/kg/day (“high dose”). This dosing protocol was maintained at the same time of day ± 2 h for 28 days.

### 2.3. Plasma Pharmacokinetics

In animals fed the 115 mg/kg/day CBD dose, pharmacokinetics were assessed both on Day 1 and Day 28. Blood samples were collected before dosing and then again at 0.5, 2, 4, 8, 12 and 24 hrs. The procedure comprised isoflurane anesthesia prior to collecting 200 μl of blood sublingually into K_2_EDTA tubes. These were kept on ice until centrifuged at 10,000× *g* for 10 min at 4 °C, and the resulting plasma was stored at −80 °C until analysis.

CBD concentrations in plasma were used to produce a best fit curve, and the following pharmacokinetic data were derived for each animal using a single compartment model; (i) peak plasma concentration (C_max_), (ii) time to peak plasma concentration (t_max_); (iii) area under the curve up to 24 h (AUC_0-24_); (iv) the absorption rate constant (K_a_) and (v) the elimination rate constant (K_e_).

### 2.4. Surgical Procedures

Following an overnight fast, animals were euthanized using CO_2_ asphyxiation, which was immediately followed by a collection of tissue samples. At least 150 mg of tissue was collected from the following sites: mid-lateral gastrocnemius of the upper leg (without any tendon tissue), the mid portion of the liver’s medial lobe; the epididymal fat in males, and periovarian fat in females. The wet weight of the tissue samples was recorded before immediately freezing at −80 °C. 

### 2.5. Validation of Tissue CBD Measurement

Tissue CBD concentrations were assessed using ultra high performance liquid chromatography with mass spectrometer detection. Multiple steps were used to validate the CBD assay for use on tissue samples. CBD calibration standards were prepared using 1.0 mg/mL CBD (Sigma-Aldrich, USA; batch No. SLCC9048). This was dissolved in rat plasma containing K_2_EDTA (Bio-IVT, Westbury, NY, USA) to give final concentrations of 5, 10, 40, 100, 400, 1000, 3000, 4500 and 5000 ng/mL. These standards were used to generate a calibration curve, which was linear across the full concentration range (r > 0.995), and across all runs there was a bias of −1.15 to 1.21%, with a CV of 5.21 to 7.68%. Internal calibration standards were prepared using 1.0 mg/mL Cannabidiol-d_3_ (Cerilliant, Round Rock, TX, USA; batch No. FE12121902). This was dissolved in rat plasma containing K_2_EDTA (Bio-IVT, USA) to give final concentrations of 5, 15, 200, 2500 and 4000 ng/mL. These standards were used to generate a linear internal standard calibration curve (r > 0.995), and across all runs there was a bias of −7.80 to 0.73%, with a CV of 0.96 to 4.2%. “Matrix blank” samples were comprised of rat plasma with K_2_EDTA (Bio-IVT, USA) with no analyte or internal standard. Running these samples confirmed that at the limit of detection neither CBD or Cannabidiol-d_3_ were present in the rat plasma supplied by (Bio-IVT, USA).

Samples of rat plasma with K_2_EDTA (Bio-IVT, USA) were spiked with Cannabidiol-d_3_ to give a final concentration of 4 ng/mL and simultaneously spiked with CBD to give final concentrations of 0.3, 50 and 80 ng/mL. These were used to confirm that the internal standard did not interfere with CBD measurement at a concentration of 4 ng/mL. For CBD standards at the lower limit of 5 ng/mL, the respective CV and bias were 5.0% and −1.1%, with these values being 4.8% and −3.0% for the upper limit of 500 ng/mL. For tissue samples, the inter assay CV was in the range of 4.0 to 6.5% with a respective bias of 2.6% and 0%. The absolute recovery of Cannabidiol-d_3_ ranged from 66 to 68% with a CV of 7.1%, while the absolute recovery of CBD ranged from 88 to 93% with a CV of 6.3%. This resulted in a normalized recovery of CBD in tissue samples between 133 and 139% with a CV of 2.7%.

### 2.6. Measurement of CBD in Tissue Samples

Tissue samples were thawed at room temperature in 15 mL centrifuge tubes, which also included around 50 Lysing Matrix D beads (1.4 mm diameter) and one to two 6 mm ceramic spheres. To each sample tube, 2 mL of phosphate buffered saline (PBS) was added before being placed on a Geno Grinder 2010 and homogenized until the tissue PBS mix became viscous. An additional 3 mL of PBS was added to each tube and homogenized until the tissue cells were fully dispersed.

Prepared tissue samples were first thawed and then vortex mixed. 20 μl of each prepared tissue or CBD calibration standard was pipetted into separate 2 ml wells of a 96-well plate. To each sample or standard, 80 μl of internal standard solution was added, comprising 50 ng/mL Cannabidiol-d_3_ in methanol with 10 mM ammonium bicarbonate (1:9 *v*/*v*). The wells were then sealed and vortex mixed for 1 min at 2000 g, before adding 300 μl of 0.5% furfuryl alcohol in acetonitrile to each well and then vortex mixing for 5 min. The plate was then centrifuged at 4500× *g* for 10 min and 120 μl of the supernatant was removed and added to a 96-well plate. The plate wells were then sealed and vortex mixed for 5 min, and the plate was centrifuged at 2000× *g* for 1 min. Preparation of the tissue samples to the point of injection onto the column were performed at room temperature.

The extracted samples were analysed for CBD using a Shimadzu LC-30AD UHPLC, with an XBridge BEH C18, 2.5 μm, 50 mm × 2.1 mm column (Xbridge, San Jose, CA, USA), maintained at 40 °C. The injection volume was 20 μl and a mobile phase gradient was used to separate CBD which comprised 0.1% formic acid in water (Mobile phase A) and 0.1% formic acid in acetonitrile in water (Mobile phase B) with a flow rate of 0.6 ml/min. The mobile phase gradient comprised 40% mobile phase B at 0.5 min, increasing to 95% mobile phase B at 4.0 min. This was maintained until 5.0 min, before returning to 40% mobile phase B at 5.2 min, resulting in a total run time of 6 min. CBD was detected using an AB Sciex 6500 Triple Quad Mass Spectrometer MS/MS detector with positive ion electrospray ionisation. The ion spray voltage was 5500 V, with a Turbo ion spray temperature of 550 °C, using nitrogen as the curtain gas, CAD gas, nebulising gas and auxiliary gas. Using these parameters, the multiple reaction monitoring transitions were 315.2 to 193.1 (dwell time 50 s) for CBD and 318.2 to 196.1 (dwell time 50 s) for Cannabidiol-d_3_. With these settings, the retention time for CBD and Cannabidiol-d_3_ was ~3 min, with excellent separation from adjacent peaks in all samples. A standard curve was generated for CBD with a lower limit of detection of 5 ng/mL and an upper limit of 5000 ng/mL. CBD and Cannabidiol-d_3_ concentrations were both determined by peak area, relative to CBD and Cannabidiol-d_3_ standards. The reported CBD values follow normalisation for recovery of the Cannabidiol-d_3_ internal standard.

### 2.7. Additional Measurements

The body mass of each animal was assessed at least twice during the familiarisation period, on day 1 of the study and at 7-day intervals thereafter. The mass of food consumed was measured throughout the experimental period. Daily cage side observations were recorded for 5 days prior to the first CBD dosing and were continued until the end of the experiment. These observations were related to skin, fur, eyes and mucus membranes, occurrence of secretions and excretions, and autonomic activity (including lacrimation, piloerection, pupil size and unusual respiratory patterns). Subjective changes in gate, posture, responses to handling, as well as clonic or tonic movements and bizarre behavior were also recorded.

### 2.8. Statistics

The study was designed to determine if (i) there was a dose response effect regarding the accumulation of CBD in tissue samples, relative to the control; (ii) if there are gender differences in the dose response, (iii) determine if changes in CBD concentrations in one tissue were associated with changes in other tissues, (iv) to determine if there are any associations between plasma pharmacokinetic parameters and changes in tissue CBD levels and (v) to determine if there were gender differences in the associations between plasma pharmacokinetics and tissue CBD accumulation. To address these questions, a multivariate analysis of variance of variance (MANOVA) was employed with three outcome variables (fat, muscle and liver) and two co-variates (groups and gender with interaction) were conducted to assess global differences between genders for tissue CBD responses. A two-factor analysis of variance (ANOVA) with interaction between gender and dose groups was conducted to measure global differences in CBD between genders in adipose, muscle and liver at day 28. When significant differences were detected between groups, pairwise *t*-tests between genders at each dose level were conducted and the corresponding *p*-values were adjusted using the Holm method.

Pairwise Pearson’s product–moment correlations were used to investigate the strength of relationship in change in CBD levels between adipose, liver and muscle tissues.

The association between the pharmacokinetic data C_max_, t_max_, AUC_0-24_, K_a_ and K_e_ on Day 1 and tissue CBD accumulation were investigated using linear regression models. These were fitted using the change in CBD in adipose, muscle and liver as dependent variables and the pharmacokinetic data on day 1 as the main independent variables, with gender as an additional covariate. An identical approach was used to evaluate the relationship between pharmacokinetic parameters on Day 28 and tissue CBD accumulation.

Data are presented as means ± SEM, and the alpha level for statistical significance was set at *p* < 0.05. To facilitate interpretation of the data, confidence intervals (CI) are reported when *p* > 0.01.

## 3. Results

All animals successfully completed the experimental interventions. Body mass changes between Day 1 and Day 28 were non-significant for males and females, both within and across groups. In addition, there were no significant differences in food intake within or across groups over time. Cage observations did not reveal behavioral changes over the course of the study in any of the groups.

Plasma pharmacokinetics were evaluated on Day 1 and Day 28, and pharmacokinetic profiles for males and females are shown in Figure 1 and Figure 2, respectively. The results of single compartment modeling of the plasma data are shown in Table 1, and modeled data was used in all subsequent statistical analyses.

Pharmacokinetic parameters are shown in Table 1, and significant correlations were observed between individual parameters as a mixed gender group, both on Day 1 and Day 28. These were K_a_ and t_max_ (r = −0.98, *p* < 0.001), K_a_ and K_e_ (r = 1, *p* < 0.001) and K_a_ and AUC_0-24_ (r = −0.57, *p* < 0.05; CI −0.86, 0.01). When split by gender, there were significant correlations between K_a_ and t_max_ (r = −0.98, *p* < 0.001) and K_a_ and K_e_ (r = 1.00, *p* < 0.001) for both females and males on Day 1. On Day 28, there were still significant correlations between K_a_ and t_max_, together with K_a_ and K_e_ for females (both r = −1.00, *p* < 0.001); however, in males only, K_a_ and K_e_ were significantly correlated (r = −1.00, *p* < 0.001). As a mixed gender group, there were no significant changes in pharmacokinetic parameters between Day 1 and Day 28 of CBD feeding. However, males showed a significant reduction in t_max_ between Day 1 and Day 28 (*p* < 0.05; CI 0.13, 1.12). During the same time period, females showed an increase in C_max_ (*p* < 0.05; CI −1084, −103). In addition, at 28 Days AUC_0-24_ was significantly greater in females than in males (*p* < 0.05; CI −43,599, −317).

In response to CBD feeding, a similar pattern of accumulation was seen in all three tissues (Table 2), with higher values in adipose tissue than muscle or liver. In adipose, CBD was elevated by both the medium and high doses (respectively, *p* < 0.05; CI 5.73, 226.61 and *p* < 0.05; CI −1.84, 219.04). In muscle and liver, CBD was elevated in response to both the medium and high doses (Table 2). 

The correlations in CBD levels across the different tissues are shown in Table 3. There were “moderate” correlations (r > 0.3) between CBD levels for adipose and muscle (*p* < 0.05, CI 0.03, 0.54) and adipose and liver (*p* < 0.01); with a “high” correlation (r > 0.6) between CBD in muscle and liver (*p* < 0.001).

In females, there were significant correlations between CBD accumulation between muscle and adipose tissue (r = 0.49, *p* < 0.05; CI 0.11, 0.75); adipose tissue and liver (r = 0.43, *p* < 0.05; CI 0.03, 0.71) and muscle and liver (r = 0.86, *p* < 0.001). In contrast, only the correlation between adipose tissue and liver was significant for males (r = 0.62, *p* < 0.001).

Gender specific differences in tissue CBD responses following low, medium and high dosing are shown in Table 4. There was a significant difference in global tissue CBD responses between genders (*p* < 0.01, F-Test). A two-factor ANOVA did not reveal statistically significant gender differences between adipose or muscle CBD levels. In contrast, liver CBD levels were higher in females than in males (*p* < 0.01), with a significant difference between genders with the medium CBD dose (*p* < 0.05; CI −2.10, −0.48). With CBD in adipose as a response variable and with dose and gender interaction as covariates, there were no significant increases in males. Adipose tissue CBD was significantly elevated in females, but only for the medium dose (*p* < 0.01). In muscle, females showed significant increases with both medium and high CBD doses (respectively, *p* < 0.01 and *p* < 0.001); while males only showed significant increases with the high dose (*p* < 0.05; CI 0.02, 1.19). Finally, when liver CBD levels in females were measured relative to controls, significant increases were observed with the medium and high doses (both *p* < 0.001), while the changes in males were non-significant.

### Relationship between Plasma Pharmacokinetics and Tissue CBD Accumulation

With a mixed gender group, positive correlations were observed for K_a_ and K_e_ on Day 1 and adipose tissue CBD concentrations (both r = 0.73, *p* < 0.01). In this group, there was a negative correlation between t_max_ on Day 1 and adipose tissue CBD concentrations (r = −0.66, *p* < 0.05; CI −0.90, −0.14). No pharmacokinetic parameters were related to adipose tissue CBD levels in females; in contrast, K_a_ and K_e_ on Day 28 were both negatively associated with adipose tissue CBD levels in males (both r = −0.99, *p* < 0.01).

On Day 1 with a mixed gender group, there were positive correlations between K_a_ and K_e_, and muscle CBD (both r = 0.76, *p* < 0.01). The mixed gender group showed a negative correlation between t_max_ on Day 1 and muscle CBD (r = −0.82, *p* < 0.001). In males, t_max_ decreased between Day 1 and Day 28 (Table 1), and this change was positively correlated with muscle CBD levels (r = 0.83, *p* < 0.05; CI 0.05, 0.98).

In the mixed gender group, there was a negative correlation between t_max_ on Day 1 and liver CBD (r = −0.58, *p* < 0.01; CI −0.87, −0.13). In the mixed gender group, C_max_ on Day 28 was positively correlated with CBD in liver (r = 0.59, *p* < 0.05; CI 0.02, 0.97). In males, t_max_ decreased between Day 1 and Day 28 (Table 1), and the change in t_max_ was positively correlated with liver CBD levels (r = 0.82, *p* < 0.05; CI 0.03, 0.98). A summary of the significant correlations between pharmacokinetic parameters and tissue CBD concentrations is given in Table 5.

## 4. Discussion

The present study investigated tissue CBD concentrations in response to control (no CBD) and “low”, “medium” and “high” oral doses. Despite recent interest in tissue CBD concentrations [23,24,25], no study has assessed the relationship between dose and uptake into muscle, liver or adipose. In the medium dose group, pharmacokinetics were also assessed to provide initial insights into the relationship to specific parameters and tissue uptake. The disparity in pharmacokinetics between oral and intravenous (iv) administration are relevant to understanding CBD metabolism in free living humans and the efficacy of CBD products. This study provides unique insights into mammalian CBD metabolism regarding pharmacokinetics after acute and chronic intake, subsequent accumulation in tissue, with additional gender specific evaluation of these parameters. There is currently a paucity of data on oral CBD pharmacokinetics, and a recent review highlighted the need for better understanding in this field [8]. Therefore, the prima facie findings from the current investigation have value in understanding CBD metabolism in mammals. These may have direct relevance for the development of optimal oral dosing regimens for humans with regards to therapeutic interventions and dietary supplementation for health.

A single compartment model was considered most appropriate to describe CBD distribution and elimination in response to oral dosing. This model assumes the body acts as a single uniform compartment, from which CBD can both enter and leave. Although the simplest pharmacokinetic models of drug distribution involve intravenous administration, they lack the construct validity provided by oral ingestion.

Gender specific pharmacokinetic data are presented in Figure 1 and Figure 2, with modeled pharmacokinetic data in Table 1. t_max_ directly relates to the rate of CBD absorption and reflects multiple underlying processes. These include gastric motility, glomerular filtration rate, hormones and differences in hepatic enzyme activity [26,27,28,29]. In the present investigation, t_max_ was ~8 h for both male and female rats. This is consistent with human studies providing a similar relative CBD dose. For example, t_max_ was 3 hrs when an acute dose comprising 800 mg CBD (~10 mg/kg bodyweight) was given to male and female cannabis smokers [30], while Taylor and co-workers [31] reported a t_max_ of 5 hrs in healthy males and females, in response to a 6000 mg CBD bolus (~80 mg/kg). The slightly longer t_max_ in the present study may reflect the higher relative CBD dose (i.e., 115 mg/kg) and data on healthy humans supports this view [18]. Increasing the oral CBD bolus from 10 mg to 100 mg resulted in t_max_ increasing from 3 hrs to 3.5 hrs [32]. Following 28 days of CBD feeding in males, we found t_max_ was reduced from 8 hrs 25 mins to 7 hrs 47 mins, although this effect was not seen in females. Human studies involving repeated CBD administration to men and women for 7 days reported reductions in t_max_ from 5 to 3 hrs [31]. The mechanisms underpinning this rapid change in t_max_ with repeated dosing are unclear. They might be modulated by increased lipid transporter activity and/or gut blood flow, in combination with increased liver metabolism and clearance [33,34]. However, it is noteworthy that neither K_a_ or K_e_ were significantly altered in males or females following 28 days of CBD feeding (Table 1). It is also surprising that the values for K_a_ and K_e_ were almost identical (Table 1). One possibility is that the best-fit lines generated by the single compartment model may have attenuated differences between Ka and Ke. This could be explored in future studies by increasing the frequency and/or duration of blood sampling.

C_max_ values in this study were approximately double those reported in healthy men and women [31]. This may reflect the higher CBD dose provided to the animals, superior CBD bioavailability or species differences. C_max_ and AUC_0-24_ are mathematically interdependent and therefore are often closely correlated. Following 28 days of CBD ingestion, this investigation found C_max_ and AUC_0-24_ were respectively increased by 36% and 44% in females. Similar findings were observed in healthy males and females in response to 7 days oral CBD feeding at 750 mg/day; with C_max_ increasing by ~50% and AUC by ~225% [31]. However, in the present study, male rats showed the opposite response, with a 22% reduction in AUC_0-24_ after 28 days of CBD feeding.

The absorption rate constant K_a_ provides a measure of CBD’s absorption into plasma, while the elimination rate constant K_e_ (sometimes abbreviated to K_el_) relates to removal from plasma. K_a_ and K_e_ have previously been measured in response to oral CBD intake without concurrent provision of THC. The study of Williams and co-workers [18] investigated undisclosed preparations providing 30 mg of CBD and reported K_a_ values of 0.24 to 1.87 l/hr, with K_e_ values of 0.27 to 0.56 l/hr. The lower range of values for K_a_ and K_e_ are comparable to those of the present investigation. However, their findings also highlight that when the CBD dose is low (0.4 g/kg) in relation to co-consumed excipients, it has a profound effect on pharmacokinetics. Other human trials reported K_e_ values ~3 fold greater than observed in the present investigation after consuming capsules containing CBD and THC [12,32]. The lower K_e_ values in rats indicate slower CBD clearance, which could be a consequence of the greater relative CBD dose provided to the experimental animals. We found no differences in K_a_ or K_e_, between genders or in response to 28 days of CBD intake. This is surprising, especially when considering the changes in t_max_ and C_max_ we observed, which suggest modification of CBD appearance and/or removal. The finding that specific pharmacokinetic parameters are modified in responsive to regular CBD intake are consistent with limited human data [31]. One human study reported higher C_max_ and AUC cannabinoid values for females than males, after co-ingestion of THC and CBD [11]. These findings are also consistent with the findings of [35], who administered CBD with MCT to humans. The authors provided evidence for gender specific differences in cannabinoid pharmacokinetics. The present investigation found evidence for higher C_max_ and AUC values in females relative to males, but only after 28 days of CBD administration. Gender specific pharmacokinetic changes in response to repeated CBD dosing are one of the unique findings of the present investigation. These findings may be useful for developing gender specific dosing strategies to optimise tissue CBD elevation.

The present investigation showed a ~20 to 180 fold greater elevation of CBD in adipose relative to muscle or liver (Table 2). It is impossible to directly compare this finding to human research, as CBD has not been assessed in human adipose tissue. However, if THC is a suitable surrogate for CBD accumulation, then the present findings are consistent with the available human data [20,36]. Furthermore, studies in the Large White pig also show greater (~3 to 16 fold) THC accumulation in adipose, than liver or muscle [37].

Tissue responses to different CBD doses with a mixed gender group are shown in Table 2. The large SEM for CBD in adipose tissue for both males and females, regardless of dose (Table 4), is indicative of high variability between animals. As they were of very similar age, reared under identical conditions and given the same relative CBD dose, other factors appear to have influence tissue CBD accumulation. One parameter which might explain the heterogeneity in adipose tissue CBD responses is pharmacogenetic differences in metabolism. High inter-subject variability in CBD pharmacokinetics have previously been reported in humans [8] and recent research has highlighted genetic differences in human CBD metabolism [38] especially when variants of CYP2C9 are present [39].

Gender specific responses to CBD accumulation in muscle, liver and adipose tissue are shown in Figure 3a–c, with quantitative data in Table 4. For the same relative CBD dose females consistently had higher concentrations in muscle and liver than in males (Table 4). Muscle and liver tissue showed a clear dose response in CBD levels for both males and females (Table 4) and these findings may have relevance to humans. Increased CBD concentrations in the liver have the potential to inhibit metabolism of the selective serotonin reuptake inhibitor Citalopram [40]. It has recently been suggested CBD could have therapeutic value for alcohol use disorder and alcohol related liver damage, via both behavioural and biochemical mechanisms [41]. These include reduced alcohol intake, and increased hepatic resistance to inflammation and oxidative damage [41]. The present finding of increased liver CBD provides a theoretical basis to support recent proposals that CBD could exert antioxidant and anti-inflammatory effects in the liver [7,41]. Elevated skeletal muscle CBD could affect contractility, antioxidant protection and exercise recovery [42,43]; which would have implications for training and sports performance [44]. It is important for future studies on the potential benefits of elevated CBD in liver, muscle and adipose employ relevant biomarkers and clinical endpoints.

In adipose tissue there was evidence for a dose response in males (Table 4), but not when considered as a mixed gender group (Table 2). Adipose tissue CBD levels with the medium and high doses were ~100 fold greater than the elevation in muscle or liver, regardless of gender (Table 4). The factors governing the transport, metabolism and accumulation of ingested CBD in tissue are extremely complex. Fatty acid binding proteins (FABP) play a key role in intracellular CBD transport; specifically FABP_3_, FABP_5_ and FABP_7_ [45]. In the extracellular compartment albumen is in the main CBD carrier, with 90% of the CBD being protein bound. The liver is considered the main site of CBD metabolism [40], with extrahepatic metabolism in the brain, intestines and lung [46]. Cytochrome p450 mediated THC metabolism involves glucuronidation to form water-soluble adducts and there is evidence CBD is transported and metabolized via similar pathways [45]. Correlations between CBD levels in adipose and muscle (r = 0.31) and adipose and liver (r = 0.37) were much lower than for muscle and liver (r = 63) (Table 3). This is a surprising finding when considering the importance of the liver in CBD metabolism [40].

The relationship between plasma pharmacokinetics and tissue CBD accumulation were explored to provide insights into CBD metabolism. With a mixed gender group, there was a negative correlation between t_max_ on Day 1 and adipose tissue CBD uptake. This means the shorter the time to reach C_max_ the greater the uptake into adipose tissue. The mixed gender group also showed relationships between K_a_ and K_e_ on Day 1 and adipose tissue CBD levels; such that the higher the rate constant for appearance and disappearance the greater the rise in adipose tissue CBD. In contrast, by Day 28 the relationship between K_a_ and K_e_ on Day 28 and adipose tissue CBD was negative, but only in males. This indicates the lower the rate constants for CBD appearance and disappearance, the higher the adipose tissue concentration. The changes in the relationship between K_a_ and K_e_, both as a mixed gender group and for males and females separately, demonstrate these pharmacokinetic parameters are not static. One rationale for this is that K_a_ and K_e_ may be modified in response to elevated CBD levels. Potential mechanisms might involve increased liver CBD metabolism and/or enhanced excretion of CBD and CBD adducts. When CBD pharmacokinetics and adipose tissue accumulation were considered solely for females no significant relationships were observed. This may indicate that ovarian hormones have a profound influence on CBD uptake into adipose tissue.

In the mixed gender group, muscle CBD was positively correlated to both K_a_ and K_e_ on Day 1. This means the higher the rate constants for absorption and elimination the greater the level of CBD in muscle, and is identical to the relationship found in adipose tissue. The mixed gender group also showed a close negative correlation between t_max_ on Day 1 and muscle CBD (r = −0.82, *p* < 0.001). So, the shorter t_max_ on Day 1, the higher the muscle CBD content on Day 28. In males t_max_ decreased between Day 1 and Day 28 (Table 1) and this change in t_max_ was positively correlated with muscle CBD levels (r = 0.83, *p* < 0.05; CI 0.05, 0.98). So, the more t_max_ decreased the greater the CBD uptake to muscle.

Liver CBD was negatively associated with t_max_ on Day 1 and positively correlated with C_max_ on Day 28 in the mixed gender group. The former finding shows that peak circulating CBD concentrations are associated with higher liver CBD. In males, the reduction in t_max_ between Day 1 and Day 28 was positively correlated with liver CBD levels. Therefore, the greater the rise in liver CBD the greater the reduction in time taken to reach peak plasma concentrations.

As C_max_ and AUC_0-24_ are key determinants of tissue CBD exposure, we anticipated significant positive correlations between C_max_ and AUC_0-24_ on Day 1 and tissue CBD. Surprisingly, neither AUC_24_ on Day 1 or Day 28 were correlated with CBD uptake into any tissue. C_max_ was only correlated with uptake into liver tissue on Day 28, and even then only within the mixed gender group.

It is important to recognise that the relationship between pharmacokinetic parameters and tissue CBD levels appear to be dynamic. One example of this is t_max_, which on Day 1 in a mixed gender group was negatively associated with CBD levels in adipose, muscle and liver. This subsequently changed so that there was no association between t_max_ on Day 28 and CBD in any tissue. Similarly, pharmacokinetic measurements that had no predictive value regarding tissue CBD uptake on Day 1, did predict tissue CBD levels when re-assessed on Day 28. For example, there was a positive association between C_max_ on Day 28 and liver CBD levels. The statically significant associations between pharmacokinetics and tissue CBD levels in the mixed gender group are outlined in Table 5.

This research trial illustrates that the relationships between CBD pharmacokinetics and tissue levels are extremely complex. However, our findings suggest plasma pharmacokinetics could be used to predict CBD accumulation in some tissues. To improve the predictive accuracy of such measures in humans, factors such as gender, prior CBD exposure, age, diet and physical activity levels should also be considered [47,48]. The use of pharmacokinetics to predict tissue CBD levels has several advantages over biopsies. These include less invasive procedures for patients and less costly procedures for experimenters. Understanding tissue CBD responses to different dosing regimes and carriers could be important in clinical and research settings. For example, in attaining therapeutic CBD concentrations in target tissues and/or facilitating the development of CBD dosing strategies for specific populations.

The present study revealed that in a mixed gender group, t_max_ is the best predictor of tissue CBD levels in response to repeated dosing. It should be noted that these parameters are negatively correlated, such that a shorter t_max_ indicates higher tissue CBD. The relationship between t_max_ and tissue CBD uptake has not been assessed in humans. If similar responses were observed they could be valuable for identifying CBD sensitive individuals (CBD responders and non-responders). Such information could be used to personalize oral CBD dosing and maximise therapeutic benefits.

The accumulation of CBD in tissues has the potential to exert antioxidant and anti-inflammatory effects [7,41], in addition to amplifying the effects of other drugs [40,49]. The current findings provide clear evidence that CBD accumulates in adipose, liver and muscle tissue; furthermore, for any given dose there is greater elevation in adipose tissue than muscle or liver.

## 5. Conclusions

The animal model of oral CBD consumption evaluated in this study produces pharmacokinetic responses that are consistent with oral CBD intake by humans. In addition, we observed the same pattern of CBD elevation in the adipose, muscle and liver of rats that occurs with THC in humans. For the same relative CBD dose, females consistently showed higher levels in muscle and liver and this relationship was also present in adipose with the low and medium CBD doses. Some pharmacokinetic parameters can predict tissue CBD levels; however, there are important gender differences in these responses. Regular CBD intake modifies some pharmacokinetic parameters and their association with tissue CBD concentrations.

Additional work on human CBD pharmacology is necessary to gain insights into its relationship to tissue CBD uptake, this work being particularly important for the dose ranges that are used therapeutically. Research into the long-term effects of oral CBD consumption over several months or years is also warranted. There should also be a specific focus on understanding CBD pharmacokinetics and tissue accumulation in conditions where CBD is commonly consumed, such as epilepsy and addiction disorders. This work will help to extend understanding of the interactions between tissue CBD levels and the drugs most commonly used in these clinical conditions.

## Figures and Tables

**Figure 1 nutrients-14-02101-f001:**
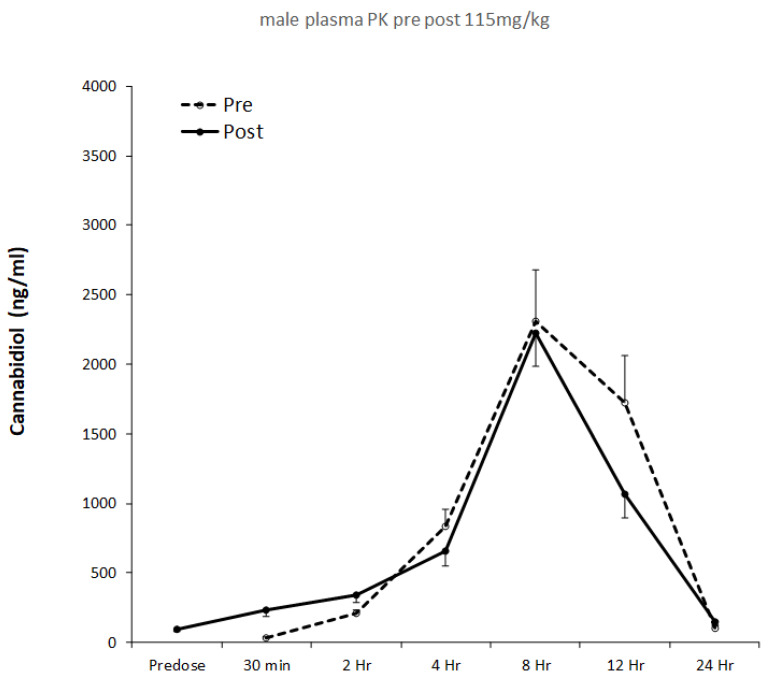
Plasma CBD pre- and post-28 days oral feeding at 115 mg/kg in males.

**Figure 2 nutrients-14-02101-f002:**
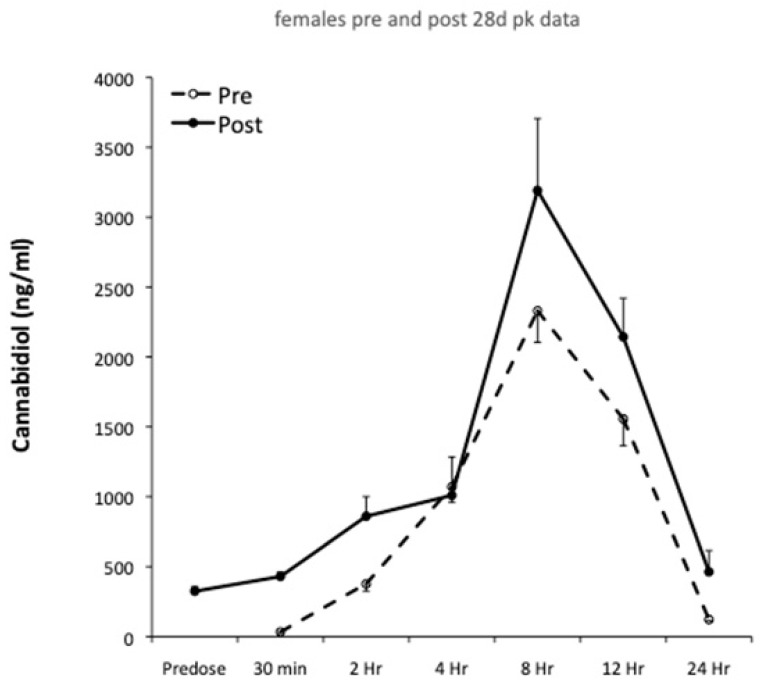
Plasma CBD pre- and post-28 days oral feeding at 115 mg/kg in females.

**Figure 3 nutrients-14-02101-f003:**
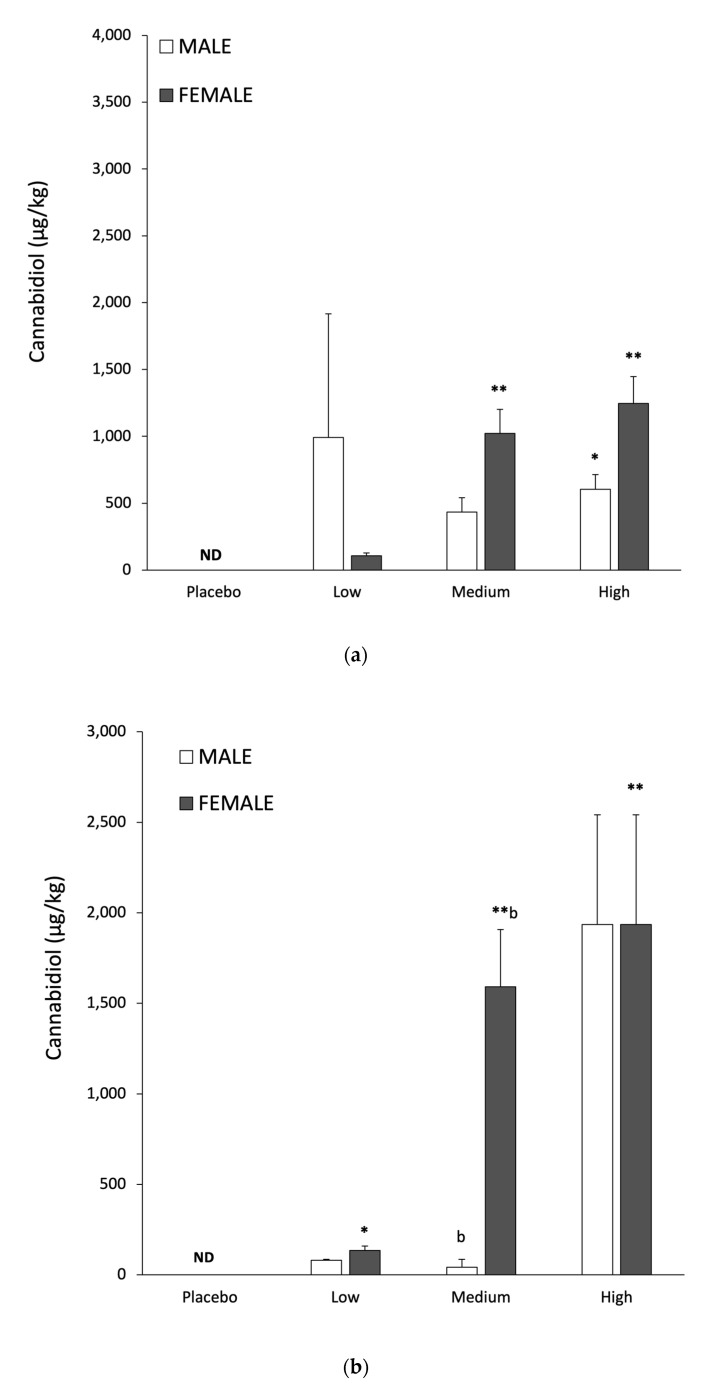
Male and female muscle (**a**), liver (**b**) and adipose (**c**), tissue CBD concentrations following 28 days feeding at low (30 mg/kg), medium (115 mg/kg), high (230 mg/kg) CBD dosing. Data are means ± SEM, * *p* < 0.05 and ** *p* < 0.01 relative to control; b = *p* < 0.01 dose specific difference between genders. ND = not detectable.

**Table 1 nutrients-14-02101-t001:** Pharmacokinetic parameters in males and females on Day 1 and Day 28.

Time Point	Gender	Pharmacokinetic Parameters				
		t_max_ (hrs:min)	C_max_ (ng/mL)	AUC_0-24_ (hr × ng/mL)	K_a_ (L/hr)	K_e_ (L/hr)
Day 1	Female	7:45 ± 0:46	1648 ± 330	35,040 ± 8449	0.13 ± 0.01	0.13 ± 0.01
	Male	8:25 ± 0:32	1588 ± 594	36,455 ± 13,521	0.12 ± 0.01	0.12 ± 0.01
Day 28	Female	8.17 ± 0:59	2242 ± 529 ^a^	50,561 ± 15,090 ^b^	0.12 ± 0.01	0.12 ± 0.01
	Male	7:47 ± 0:16 ^a^	1350 ± 388	28,602 ± 8486 ^b^	0.13 ± 0.00	0.13 ± 0.00

Data are means ± SEM, ^a^ within group difference, ^b^ difference between groups.

**Table 2 nutrients-14-02101-t002:** The effects of CBD dosing on 28-day tissue concentrations with mixed gender groups.

Dose	Tissue CBD Concentration (mg/kg)		
	Adipose	Muscle	Liver
Low (30 mg/kg/day)	5.30 ± 4.24	0.27 ± 0.25	0.07 ± 0.03
Medium (115 mg/kg/day)	116.17 ± 61.27 *	0.64 ± 0.17 **	0.95 ± 0.18 **
High (230 mg/kg/day)	108.60 ± 86.34 *	0.93 ± 0.31 ***	1.15 ± 0.34 ***

Data are means ± SEM, No CBD was detected in control tissue. * *p* < 0.05; ** *p* < 0.01; *** *p* < 0.001 relative to control.

**Table 3 nutrients-14-02101-t003:** The pairwise correlations between CBD levels across tissues with mixed gender groups.

Tissue	r-Value
Adipose and Muscle	0.31 *
Adipose and Liver	0.37 **
Muscle and Liver	0.63 ***

* *p* < 0.05; ** *p* < 0.01, *** *p* < 0.001.

**Table 4 nutrients-14-02101-t004:** The gender differences in tissue responses with different CBD doses.

Dose	Tissue CBD Concentration (mg/kg)					
	Adipose		Muscle		Liver ^a^	
	Female	Male	Female	Male	Female	Male
Low (30 mg/kg/day)	7.79 ± 7.29	2.79 ± 1.20	0.04 ± 0.02	0.50 ± 0.47	0.09 ± 0.03 *	0.05 ± 0.02
Medium (115 mg/kg/day)	197.01 ± 101.52 *	35.31 ± 21.01	0.85 ± 0.22 **	0.43 ± 0.11	1.59 ± 0.31 **^,b^	0.30 ± 0.04 ^b^
High (230 mg/kg/day)	84.24 ± 32.97	132.95 ± 106.74	1.25 ± 0.20 **	0.60 ± 0.11 *	1.94 ± 0.60 **	0.37 ± 0.07

Data are means ± SEM; ^a^ *p* < 0.05 group difference between males and females * *p* < 0.05 and ** *p* < 0.01 relative to control; ^b^ *p* < 0.01 dose specific difference between genders.

**Table 5 nutrients-14-02101-t005:** The summary of significant correlations between pharmacokinetics and tissue CBD levels with a mixed gender group.

Pharmacokinetic Parameters	Tissue					
	Adipose		Muscle		Liver	
	Day 1	Day 28	Day 1	Day 28	Day 1	Day 28
t_max_	−0.66 *		−0.82 ***		−0.58 **	
C_max_						0.59 *
K_a_	0.73 **		0.76 **			
K_e_	0.73 **		0.76 **			

The numbers denote the r value, * *p* < 0.05; ** *p* < 0.01; *** *p* < 0.001.

## Data Availability

Not applicable.

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
