# Peer review of "Cannabidiol (CBD) Dosing: Plasma Pharmacokinetics and Effects on Accumulation in Skeletal Muscle, Liver and Adipose Tissue"

_nutrients, 2022, doi:10.3390/nu14102101_

Round 1

Reviewer 1 Report

The subject of the present paper is of interest if we consider the potential use of CBD for the treatment of different diseases, as cited by the authors. For this, CBD pharmacokinetics must be better understood in order to optimize its formulation.  

The paper reports a lot of interesting information. However, the main comment that would need to be made is on the quantification of CBD in the samples. As all the paper data and conclusions are based on the quantification of CBD in the different biological samples, more attention should be put on the analytical approach used to perform the quantification. If LC-MS method used, what are the mass parameters? How was the CBD quantified (m/z for quantification, identification)? A detailed description of the method is missing. Indeed, sample preparation is described but not enough information on the analytical part is communicated. For accurate measurement, a proper method validation and communication of validation parameters should be made. If the authors have used a previously described method, appropriate reference should be mentioned. 

Additional comments and suggestions:

- L154-155: What is the nature and concentration of the internal standard solution mentioned? Is it the one described L192? May be it should be mentioned earlier in the paper, in the material and methods.

- L158-159: could the authors clarify why there were different volumes (120µL of the supernatant and 300µL for the solvent blank)?

- L161-162 : What are LLOQ samples? Could the authors rephrase to allow the reader to better understand the sentence and the experimental approach?

- L173 : QC samples are mentioned. Could the authors describe how they were prepared (levels of CBD in these samples, done for the different matrices)? If they are mentioned I would recommend to better describe them.

- L183-184: could the authors explain why different volumes, as on L158-159 (120µL of samples and 180µL of water) and why water in place of blank instead of solvent?

- L188: the quantification of the CBD in tissue samples was done by LC-MS. A lot of information regarding the method used are missing: the mass parameters used for the quantification and identification, positive or negative mode, the m/z … Could the authors provide information regarding MS analysis of these samples. Or please refer to another paper in which the method is clearly described.  

Was the analytical method validated for the targeted matrices? Is the uncertainty of the method known? What is the recovery of the CBD extraction from the different matrices? How can the authors be sure about the robustness of the quantification of CBD in the different samples?

- L292, Table 1: Ka and Ke should be expressed in 1/hr (hr-1)?

- L292, Table 1: Isn’t it surprising to obtain exact same numbers for Ka and Ke? Could the authors discuss this point?

- L300, L308, 311: shouldn’t the CI of AUC be expressed in %?

- Table 2: the SEM for CBD content in the 3 tissues are the same for the 3 different doses (low, medium, and high). How can they be the same for 3 different groups of dose/animals?

- Table 5: if there is no correlation showed for AUC, the line should be removed from the Table. Why to show only the significant data?

- Figure 3: a, b, c are missing and legend on vertical axis not properly appearing.

- From L455 to 478: the authors compare their current data with literature, based on the dose of CBD administered. It would be interesting to also mention the formulation of CBD for all the cited references in addition to the doses, in order to have an idea of the potential influence of the formulation (how it is administered to the subjects, capsules, cigarettes, presence of excipient, presence of oil…) on the absorption.

- Only CBD was measured, no metabolite. The conclusions could be different if all metabolites had been measured. This aspect could be discussed.

Author Response

General comments to the reviewers

The reviewers raised a number of points regarding the methods requesting more detail about the analysis and clarification of the results. In addition the reviewers also highlighted several points that warranted more detailed analysis and discussion. Addressing the points raised by the reviewers has significantly improved the manuscript and the authors are extremely grateful for the insights they provided.

Reviewer comments

General

How was the CBD quantified (m/z for quantification, identification)? A detailed description of the method is missing. Indeed, sample preparation is described but not enough information on the analytical part is communicated. For accurate measurement, a proper method validation and communication of validation parameters should be made. If the authors have used a previously described method, appropriate reference should be mentioned.

The manuscript has been revised to include a section on method validation (section 2.5) and detail sample analysis (section 2.6)

Reviewer 1 specific comments

What were the mass parameters used for Mass Spectrometer detection of CBD?

For tissue CBD concentrations, positive ionization with multiple reaction monitoring was used to monitor the ion transition from m/z 315.2 to m/z 193.1. For the internal standard the ion transition parameters were m/z 318.2 to m/z 196.1.  This information is now included in the revised manuscript (Lines 222 to 223).

Was the Mass Spectrophotometer in ‘positive’ or ‘negative’ mode?

It was in positive mode, which is now stated in the revised manuscript (Lines 219 to 220).

What was the M/Z ratio for CBD quantification and identification?

The m/z ratios for tissue CBD and the internal standard are now included in the revised manuscript (Lines 223 to 224)

Was the Mass Spectrometer method to detect CBD based on a published method that we could refer to?

It is a validated but unpublished method. We have now included a section on method validation (Lines 153 to 186) 

Could you provide details on the validation of the CBD used, and the parameters used to establish validation?

We have now included a section on method validation (Lines 153 to 186)

L154-155: What is the nature and concentration of the internal standard solution mentioned? Is it the one described L192? May be it should be mentioned earlier in the paper, in the material and methods.

Details of nature and concentration of the internal standard are included earlier in the paper, in section 2.5 on validation of tissue CBD measurement. 

L158-159: could the authors clarify why there were different volumes (120µL of the supernatant and 300µL for the solvent blank)?

The methods used for assay validation and the measurement of CBD in tissue samples are now clearly explained in the revised manuscript in sections 2.5 and 2.6

L161-162 : What are LLOQ samples? Could the authors rephrase to allow the reader to better understand the sentence and the experimental approach? Explain what LLOQ samples are and rephrase

Reference to ‘LLOQ’ was a typographical error and has been removed from the manuscript. 

L173 : QC samples are mentioned. Could the authors describe how they were prepared (levels of CBD in these samples, done for the different matrices)? If they are mentioned I would recommend to better describe them.

The methods used for assay validation and the measurement of CBD in tissue samples are now clearly explained in the revised manuscript in sections 2.5 and 2.6

L183-184: could the authors explain why different volumes, as on L158-159 (120µL of samples and 180µL of water) and why water in place of blank instead of solvent?

The methods used for assay validation and the measurement of CBD in tissue samples are now clearly explained in the revised manuscript in sections 2.5 and 2.6

L188: the quantification of the CBD in tissue samples was done by LC-MS. A lot of information regarding the method used are missing: the mass parameters used for the quantification and identification, positive or negative mode, the m/z … Could the authors provide information regarding MS analysis of these samples. Or please refer to another paper in which the method is clearly described.

The methods used for assay validation and the measurement of CBD in tissue samples are now clearly explained in the revised manuscript in sections 2.5 and 2.6

Was the analytical method validated for the targeted matrices? Is the uncertainty of the method known? What is the recovery of the CBD extraction from the different matrices? How can the authors be sure about the robustness of the quantification of CBD in the different samples?

The method was validated for the target matrices and the revised manuscript now includes details of how the method was validated and the associated performance data. 

Is the uncertainty of the method known ?

Yes – we have included data on method uncertainly in section 2.5 of the revised manuscript.

How can the authors be sure about the robustness and quantification of CBD in the different samples?

Assay validation was performed on CBD measurement in tissue samples, details of this work and the performance of the assay are now included in section 2.5 of the revised manuscript.

L292, Table 1: Ka and Ke should be expressed in 1/hr (hr-1)

The manuscript has been revised so that the units for Ka and Ke are expressed in l/hr throughout

L292, Table 1: Isn’t it surprising to obtain exact same numbers for Ka and Ke? Could the authors discuss this point?  

The numbers for Ka and Ke are very similar but not identical. The absorption rate constant is higher than the elimination rate constant to a certain time (prior to the peak plasma concentration); then, the rate of absorption will be equal to the rate of elimination. After which elimination exceeds absorption. In a single compartment pharmacokinetic model a line of best fit is applied to the available data points.  In this study we have circa 3 points over what maybe considered the terminal end of the plasma curve (so peak to last data measurement).  However, this did not reach zero in all animals, therefore the best fit line generated may have attenuated the small differences between Ka and Ke. This point is highlighted in the revised manuscript (Lines 464 to L469).

L300, L308, 311: shouldn’t the CI of AUC be expressed in %?

The authors are not clear on what the reviewer means by expressing CI as a percentage. It is possible to divide the two ends of CI by the estimated difference between females and males, to give you a CI expressed in terms of percentages of the estimated difference. This is not the conventional way that CI data is presented.

Table 2: the SEM for CBD content in the 3 tissues are the same for the 3 different doses (low, medium, and high). How can they be the same for 3 different groups of dose/animals?

We are very grateful to the reviewers for picking up on this. The mean values were correct, but the SEM were incorrect and have now all been revised.

Table 5: if there is no correlation showed for AUC, the line should be removed from the Table. Why to show only the significant data?

As suggested by the reviewer the AUC line has been removed from Table 5. Only the significant data are presented as it provides a concise way to present the information to the reader.

Figure 3: a, b, c are missing and legend on vertical axis not properly appearing.

Figure 3 a, b and c has been redrawn and now all the axis should appear correctly.

From L455 to 478: the authors compare their current data with literature, based on the dose of CBD administered. It would be interesting to also mention the formulation of CBD for all the cited references in addition to the doses, in order to have an idea of the potential influence of the formulation (how it is administered to the subjects, capsules, cigarettes, presence of excipient, presence of oil…) on the absorption.

The present investigation was specifically focused on oral CBD pharmacokinetics and tissue accumulation. Millar et al. 2018 reviewed CBD pharmacokinetics with different routes of CBD administration, as proposed by the reviewer and is cited in the manuscript. Other methods of CBD delivery such as oral dermal inhalation can have therapeutic benefits and there are gaps in existing literature regarding different routes of administration and delivery vehicles that undoubtedly warrant further investigation.

Only CBD was measured, no metabolite. The conclusions could be different if all metabolites had been measured. This aspect could be discussed.

The focus in the present investigation was to assess CBD accumulation in muscle liver and fat. CBD excretion and metabolism was not assessed. The accumulation of CBD metabolites in tissue would certainly be interesting to look at in future work.

Reviewer 2 Questions and comments

Line 40. According with nomenclature “Cannabis sativa L” should be written in italicsCannabis sativa L

Response - This has now been written in italics.

Line 72. Check whether “substances” should be written on singular noun “substance”

This has been edited to “substance’s”

Line 210. It appears that a closing bracket was missed: “... patterns).”

Response - The second bracket has now been included

Line 360. Assess whether the font size and underlining are wright.

Response – The font size has been changed and the underlining removed to make it consistent throughout the manuscript.

Line 369. To be consistent with other table text headers, Table 5 text header should not be underlined.

Agreed. This is no longer underlined.

Lines 406 - 421. Figure 3 appears incomplete, 3b and 3c figures are split on the top part. This must be amended. In addition, it should be stated a, b and c close to each graphic panel, to identify them and avoid any misinterpretation.

The figures have been edited to make interpretation clearer

Lines 546-548. The sentence “Increased CBD concentrations in the liver are likely to affect drug metabolism [40]...” should be clarified. How do CBD concentrations affect drug metabolism? increasing (positively) or decreasing (negatively)?

CBD has the potential to inhibit metabolism of the selective serotonin reuptake inhibitor Citalopram.  This is now stated in the manuscript (Lines 537 to L538).

It is important to argue and state it to justify the following sentence “... and it has recently been suggested CBD could have therapeutic value for alcohol use disorder and alcohol related liver damage [41].”

The manuscript has been updated to include some potential mechanisms to support this statement (Lines 539 to L543).

Consider improve the quality of figures. Edition in colour may help.

The figures have been updated to make them clearer.  

Reviewer 2 Report

Oral Cannabidiol (CBD) dosing: plasma pharmacokinetics and effects on accumulation in
skeletal muscle, liver and adipose tissue
Dear Authors,
Assess and amend the following minor defects:
Line 40. According with nomenclature “Cannabis sativa L” should be written in italics
“Cannabis sativa L”
Line 72. Check whether “substances” should be written on singular noun “substance”
Line 210. It appears that a closing bracket was missed: “... patterns).”
Line 360. Assess whether the font size and underlining are wright.
Line 369. To be consistent with other table text headers, Table 5 text header should not be
underlined.
Lines 406 - 421. Figure 3 appear incomplete, 3b and 3c figures are split on the top part. This
must be amended. In addition, it should be stated a, b and c close to each graphic panel,
to identify them and avoid any misinterpretation.
Lines 546-548. The sentence “Increased CBD concentrations in the liver are likely to affect
drug metabolism [40]...” should be clarified. How do CBD concentrations affect drug
metabolism? increasing (positively) or decreasing (negatively)? It is important to argue
and state it to justify the following sentence “... and it has recently been suggested CBD
could have therapeutic value for alcohol use disorder and alcohol related liver damage
[41].”
Consider improve the quality of figures. Edition in colour may help.

Author Response

(The authors gave the same response as above.)
